# Differential Associations of Intakes of Whole Grains and Coarse Grains with Risks of Cardiometabolic Factors among Adults in China

**DOI:** 10.3390/nu14102109

**Published:** 2022-05-18

**Authors:** Qiumin Huang, Lixin Hao, Liusen Wang, Hongru Jiang, Weiyi Li, Shaoshunzi Wang, Xiaofang Jia, Feifei Huang, Huijun Wang, Bing Zhang, Gangqiang Ding, Zhihong Wang

**Affiliations:** 1National Institute for Nutrition and Health, Chinese Center for Disease Control and Prevention, 29 Nanwei Road, Beijing 100050, China; qmhuangx@163.com (Q.H.); haolx@ninh.chinacdc.cn (L.H.); wangls@ninh.chinacdc.cn (L.W.); jianghr@ninh.chinacdc.cn (H.J.); liwy@ninh.chinacdc.cn (W.L.); wangssz@ninh.chinacdc.cn (S.W.); jiaxf@ninh.chinacdc.cn (X.J.); huangff@ninh.chinacdc.cn (F.H.); wanghj@ninh.chinacdc.cn (H.W.); zhangbing@chinacdc.cn (B.Z.); dinggq@chinacdc.cn (G.D.); 2Key Laboratory of Trace Elements and Nutrition, National Health Commission, Beijing 100050, China

**Keywords:** whole grains, coarse grains, cardiometabolic factors, China, adults

## Abstract

There is a lack of studies on the association between whole grain intake and cardiometabolic risk factors in China and the current definition of whole grains is inconsistent. This study defined whole grains in two ways, Western versus traditional, and examined their associations with the risks of major cardiometabolic factors (CMFs) among 4706 Chinese adults aged ≥18 years, who participated in surveys both in 2011 and in 2015. Diet data were collected by consecutive 3 d 24 h recalls, together with household seasoning weighing. Whole grains were defined as grains with a ratio of fiber to carbohydrate of ≥0.1, while coarse grains were defined as grains except for rice and its products, and wheat and its products. Multivariable logistic regressions were modeled to analyze the associations of intakes of whole grains and coarse grains, respectively, with risks of major CMFs including obesity-, blood pressure-, blood glucose- and lipid-related factors, which were defined by International Diabetes Federation and AHA/NHLBI criteria. After adjusting for potential confounders, the odds of elevated LDL-C decreased with the increasing intake levels of whole grains (OR 0.64, 95% CI 0.46–0.88, *p*-trend < 0.05). Moreover, adults with the whole grain intake of 50.00 to 150.00 g/day had 27% lower odds of overweight and obesity (OR 0.73, 95% CI 0.54–0.99) and 31% lower odds of elevated LDL-C (OR 0.69, 95% CI 0.49–0.96), as compared with non-consumers. In conclusion, given the significant nutrient profiles of whole grains and coarse grains, the adults with higher intakes of whole grains only may have a lower risk of LDL-C and overweight and obesity.

## 1. Introduction

The growing epidemic of cardiovascular diseases (CVDs) and related diseases has become a source of public health concern worldwide [1,2,3]. As a cluster of interrelated and modifiable risk factors for CVDs, cardiometabolic factors (CMFs) have gained more and more attention in the past few decades. They include overweight and general obesity, abdominal obesity, elevated blood pressure (BP), reduced high-density lipoprotein cholesterol (HDL-C), elevated triglycerides (TG), elevated low-density lipoprotein cholesterol (LDL-C), elevated fasting blood glucose (FBG), and insulin resistance [4,5].

It is recognized that an unhealthy diet is related to a high risk of CVDs and mortality. One recent study in China showed that suboptimal diet quality was associated with 20.8% of total mortality and a low intake of whole grains was one of the top four related risk factors [6]. The grain consumption patterns, combination profiles of grain subtypes or classification of grains play a key role in preventing chronic diseases. The miscellaneous association of grain subtype with health outcome may be due to the lack of standardized classification of grain subtypes, even for whole grains [7,8,9,10,11,12,13,14]. The current results about the effect of whole-grain intake on CMFs are mixed. Several studies showed that refined grain intake was positively associated with the risks of major CMFs [7,8,9,10], whereas whole grain was inversely associated with the risk of CMFs [11,12] and further against CVD-related outcomes [13]. In contrast, several studies have reported no such associations with either refined grains or whole grains [14]. Kissock et al. found a significant impact of whole-grain food definition on its potential association with body weight changes [15]. Even for the same single grain food, controversial associations with certain CMFs were observed, which may be due to different diet habits of each ethnicity. Several studies indicated that white rice intake was associated with serum TG and HDL-C levels in Indian and Korean populations [16,17]; however, null association was found between white rice intake and the risk of certain CMFs in Iranian populations [18]. In China, the traditional classification of grains generally consists of rice and its products, wheat products and other grains. The other grains are approximately regarded as coarse grains, similar to the Western-defined whole grains. Huang et al., found a positive association between the intake of wheat and its products and a risk of metabolic syndrome (MetS), which clusters several CMFs, while a negative relationship between rice consumption and MetS was found in southern China [19].

To date, research on the association of whole-grain intake with CMFs is lacking in China. The present study defined whole grains in two ways, Western versus traditional, and examined their associations with risks of major CMFs among 4706 Chinese adults aged ≥18 years, using dietary intake and covariates measured in the China Health and Nutrition Survey (CHNS) 2011 and major CMFs measured in CHNS 2015.

## 2. Subjects and Methods

### 2.1. Study Population

We derived all the data from the CHNS, a longitudinal cohort study about the socioeconomic transformation and its potential impacts on the nutritional status and health in China [20]. It was initiated in 1989 and then followed up every 2 to 4 years across 8 or 9 provinces (autonomous regions) between 1989 and 2009, 12 provinces in 2011, and 15 provinces in 2015 and 2018. The detailed survey design has been reported elsewhere [20]. Fasting blood samples were collected in 2009, 2015 and 2018.

Our study selected all the adults aged ≥18 years who participated in both 2011 and 2015 as the subjects, who had complete baseline data of diet, socio-demography and lifestyle in CHNS 2011, and CMFs data in 2015. We excluded those previously diagnosed or receiving treatments for diabetes, hypertension, stroke, cancer and myocardial infarction disease; those with implausible energy intakes (for men <800 or >6000 kcal/d, for women <600 or >4000 kcal/day); and pregnant or lactating women. Our final sample included 4706 adults (2108 males and 2598 females). This study was reviewed and approved by the institutional review board of the Chinese Center for Disease Control and Prevention (No. 201524) and the University of North Carolina at Chapel Hill. Written informed consent of all the participants was collected.

### 2.2. Dietary Data

The trained health workers collected diet data using consecutive 3 d 24 h (including two weekdays and one weekend day) for each individual, combined with household seasoning weighing over the same period. The details of the method are described elsewhere [20,21]. The individuals’ intake of seasonings was estimated based on the individuals’ proportion of energy intake from home-cooked foods among family members. The daily intakes of total energy, carbohydrate, sodium, and water-insoluble fiber were calculated by linking food intakes to the China Food Composition Table [19]. Total cereal fiber was calculated according to the convert index, the ratio between total fiber and water-insoluble fiber. The convert index for cereals was 1.35 [22].

### 2.3. Assessment of Intakes of Whole Grains

Based on the China Food Composition Table [23,24], the food group grains and products consist of seven subtypes including wheat, rice, corn, barley, millet, glutinous millet, others and related products. We used two methods, traditional and Western, to define whole grains. We named the traditional definition of whole grains as coarse grains, a concept relative to wheat and rice including corn, barley, millet, glutinous millet, and others. The food-based grain classification of coarse grains may have the drawback of great variation in nutrient profiles. Using the Western definition, whole grains were defined as grains with a fiber to carbohydrate ratio of ≥0.1, a concept relative to refined grains with a fiber to carbohydrate ratio <0.1, which is usually used in the Western diet [25].

Considering the extremely low consumption rate of coarse grains or whole grains, we categorized their intakes into non-consumers and tertiles among the corresponding consumers. In order to identify the intakes of coarse grains and whole grains in relation to the risks of each CMF among consumers, we further categorized them among their consumers into four levels, based on the cut-points of recommendations from the Chinese dietary guidelines, which are as follows [26]: 0.00 g/day, 0.01–≤50.00 g/day, 50.01–≤150.00 g/day, and >150.00 g/day. The lowest intake level was used as the reference group.

### 2.4. Assessment of Major CMFs

#### 2.4.1. Obesity-Related Factors

The trained workers used BC601, TANITA to measure the participants’ weight and height to the nearest 0.1 kg and 0.1 cm, respectively, who were in lightweight clothing and without shoes. Waist circumference (WC) was measured in centimeters at the midway point between the lowest rib margin and the top of the iliac crest using a SECA tape measure. Body mass index (BMI) was calculated as the measured weight (kg) divided by the height (m) squared (kg/m^2^). Overweight and obesity was defined as a BMI of ≥24.0 kg/cm [27]; abdominal obesity was defined as a WC of ≥90 cm for men and WC of ≥80 cm for women [28].

#### 2.4.2. Elevated BP

The trained workers used regularly calibrated mercury sphygmomanometers to measure the seated participants’ BP on the right arm three times in a quiet room. They read Korotkoff phase 1 (the first appearance of a pulse sound) as the systolic blood pressure (SBP) and Korotkoff phase 5 (at the disappearance of the pulse sound) as the diastolic blood pressure (DBP). We used the mean of three satisfactory measurements for the analyses. Elevated BP was defined as the mean of SBP ≥130 mmHg, or DBP ≥85 mmHg, or taking antihypertension medication [28].

#### 2.4.3. Lipid-Related Factors

Fasting blood samples were collected via venipuncture by experienced physicians, phlebotomists, or nurses. Plasma and serum samples were frozen and stored at −80 °C for later laboratory analysis. The glycerol-phosphate oxidase method was used to measure TG, and the polyethylene glycol-modified enzyme method by determiner regents for LDL-C and HDL-C measurement. We produced the following definitions: elevated TG (>1.7 mmol/L or drug treatment for elevated TG), elevated LDL-C (>3.4 mmol/L or drug treatment for elevated LDL-C), reduced HDL-C (for men <1.03 mmol/L, for women <1.29 mmol/L, or drug treatment for reduced HDL-C); lipid-related factors (having any of elevated TG, elevated LDL-C, and reduced HDL-C) [28].

#### 2.4.4. Glucose-Related Factors

FBG was measured with the glucose oxidase-phenol amino phenazone method; serum insulin was tested using a radioimmunology assay kit with the use of an XH-6020 gammacounter. Elevated FBG was defined as ≥100 mg/dL, or drug treatment of elevated FBG; HOMA-IR was defined as abnormal if HOMA-IR was >2.69 and was calculated using the following formula: fasting insulin (µU/mL) × fasting blood glucose (mmol/L)/22.5 [29,30].

### 2.5. Assessment of Covariates

Information on the sociodemographic and lifestyle variables was collected through interviewer-administered questionnaires. The baseline age was grouped as 18–44 years (y), 45–64 y, and ≥65 y; education level as less than primary, primary, and higher than primary school and the yearly income per capita was grouped into tertiles. The community urbanicity index [31] was also categorized into tertiles. Physical activity was assessed by metabolic equivalent-hours/week (MET-h/week) based on the American College of Sports Medicine Association’s standard [32]. We also considered smoking and drinking status (current smoking/drinking, and former or never smoking/ drinking) and baseline total energy intake as potential covariates.

### 2.6. Statistical Analysis

We presented the mean value and its standard error for the continuous variables and percentage for the categorical variables as a whole. We used Chi-square tests for the categorical variables and general linear regression for the continuous variables to test for differences and trends across the levels of whole grain intake (Table 1) and coarse grain intake (Appendix A). Second, the significance for the prevalence of CMFs was determined by using the Chi-square test, and by using the Cochran–Mantel–Haenszel test to test for trends across the baseline levels of whole grain intake (Table 2) and coarse grain intake (Appendix A). Third, we constructed multivariable binomial logistic regression models to assess the associations of intakes of whole grains with the risks of each CMFs, adjusting for gender, education, urbanicity index, physical activity, intakes of total energy, red meat, cooking oil, and refined grains; the same modelling method was used for coarse grains, adjusting for age, education level, urbanicity index, smoking status, intakes of total energy, tuber, red meat, poultry, fish, vegetables and fruits, cooking oil, sodium, and other grains (Table 3 and Table 4). We also tested the significance of linear trends for OR values by assigning median values to tertiles of whole grain and coarse grain intakes and modeled this variable as a continuous term. Finally, based on the definitions of the whole grains and coarse grains, we linked all the nutrient contents of each grain food included using China Food Composition Tables and estimated the average amount of each nutrient per 100 g from all the grain foods included in whole grains and coarse grains, respectively. The significance for the nutrient profiles of coarse grains and whole grains was tested using general linear regression (Table 5). All the statistical analyses were performed using SAS version 9.4 software. Formal hypothesis testing was two sided, with a significance level of 0.05.

## 3. Results

### 3.1. Basic Characteristics

As shown in Table 1, only 15.40% of the participants consumed whole grains. Moreover, the intake of whole grains among the consumers was relatively low. The consumer with the top tertile intake of whole grains had a median intake of 80.00 g/day. The adults with higher intakes of whole grains tended to be women, lived in low urbanized areas, higher physical activity levels, as well as lower intakes of red meat, vegetables and fruits, or cooking oil, but higher total energy intakes (*p* < 0.05). No significant difference in whole grain intake was found across the age groups, levels of income, education, smoking and drinking (*p* > 0.05). No disparity in the intakes of tuber, fish, poultry, and sodium was observed across the levels of whole grain intake (*p* > 0.05). 

As for coarse grains, 28.67% of the adults were consumers and the top tertile median intake of coarse grains was 116.67 g/day. Except for similarly living in low urbanized areas, the adults with higher intakes of coarse grains had quite different characteristics and tended to be smokers, aged 45–64 years, or had an education lower than primary school (*p* < 0.05). However, physical activity level, gender, income levels and drinking were not associated with intakes of coarse grains (*p* > 0.05). The adults had significantly different intakes of food groups, total energy and sodium across the levels of coarse grains (*p* < 0.05), positive association with intakes of total energy, wheat and products, tuber, vegetables and fruits, and fish, and negative association with intakes of rice and products, red meat, poultry, cooking oils and sodium (Appendix A).

### 3.2. Prevalence of Each CMFs across the Intake Levels of Whole Grains and Coarse Grains

Table 2 shows that the prevalence of only abdominal obesity, elevated BP, lipid-related risk factors and reduced HDL-C were significantly different across the baseline levels of whole grain intake in Chinese adults (*p* < 0.05). The intake of whole grains was negatively associated with the prevalence of reduced HDL-C and elevated LDL-C, but positively related with the prevalence of abdominal obesity. The adults with top tertile intake of whole grains had the lowest prevalence of elevated LDL-C (27.27%) and relatively low prevalence of reduced HDL-C (39.26%), which was higher than that of the non-consumers (35.57%).

Differently from the whole grains, a higher intake of coarse grains was linearly associated with a lower prevalence of abdominal obesity and elevated TG (*p* < 0.05). However, the consumers with the bottom tertile intake of coarse grain had the lowest prevalence of overweight and obesity, and elevated BP (46.35 and 47.29%), higher than those of the non-consumers, respectively. The intake of coarse grains was not related to the prevalence of elevated FBG and abnormal insulin resistance (*p* > 0.05) (Appendix A).

### 3.3. Associations of Coarse Grains and Whole Grain Intake with Risk of Each CMF

After adjusting for potential confounders, we observed that the prevalence odds of elevated LDL-C decreased with the increasing intake levels of whole grains (*p*-trend < 0.05), as shown in Table 3. Moreover, adults with the whole grain intake of 50.00 to 150.00 g/day had 27% lower odds of overweight and obesity (OR 0.73, 95% CI 0.54–0.99) and 31% lower odds of elevated LDL-C (OR 0.69, 95% CI 0.49–0.96), compared with that of non-consumers (Table 4). However, we did not observe any significant association between coarse grain intake and risk of each CMF here (*p* > 0.05).

### 3.4. Comparison of Nutrient Profiles of Coarse Grains and Whole Grains

As shown in Table 5, the nutrients contained in whole grains were much greater than that in coarse grains. Specifically, thiamin, niacin, vitamin E, potassium, phosphorus, manganese, zinc, and selenium in whole grains were two times higher than that in coarse grains (all *p* < 0.001), respectively. Notably, the fiber in whole grains was five times than that in coarse grains (*p* < 0.001).

## 4. Discussion

This study focused on the intakes of whole grains and coarse grains in relation to major CMFs using the CHNS data. Previous studies indicated that the consumption of coarse grains or whole grains has been associated with a lower risk of cardiovascular disease, type 2 diabetes mellitus, MetS and its components [15,16,33,34], although the definitions are inconsistent and mechanisms of these beneficial effects are not clear. Since the Chinese dietary guidelines recommend taking 50–150 g/day of whole grains and mixed legumes for adults [26], and the mixed legumes consumption of the Chinese population is too low, we referred to this cut-point to categorize the intake levels of coarse grains and whole grains. The present study showed that a whole grain intake level of 50.00 to 150.00 g/day would reduce the prevalence of overweight and obesity and elevated LDL-C, compared with the non-consumer groups, which might suggest that a moderate intake of whole grains might reduce the risk of some CMFs. We also observed significant linear trends between whole grain intake levels and elevated LDL-C, while similar results were not observed for coarse grains, which might result from their different levels of main nutrients. We learn that coarse grains and whole grains may have a great different impact on the major CMFs. The consumption rates of both whole grains and coarse grains in China is quite low and their intakes were far from the recommendations, even among the consumers [26]. In addition, the varieties of whole grains in China are not rich as those in the Western countries, which may limit the abilities of evaluating their relationship with the risk of each CMF.

Based on the traditional subtypes of grains, namely rice and products, wheat and products, and coarse grains, the results for the risks of CMFs may be very different from Western whole grain-related categorization. It was found that both intakes of rice and its products, and wheat and its products were positively associated with the prevalence of elevated TG; the same associations were observed for refined grains (data not shown here). These findings were similar to the results of previous studies conducted in Chinese [35], Korean [36], and urban Asian Indian populations [10]. Although the Jiangsu Nutrition Study showed that a higher consumption of rice (>401 g/day) was associated with less weight gain (−2.08 kg, 95% CI: 2.75–1.41), compared to the rice consumption of <200 g/day in a five-year follow-up [37], most of the previous studies showed that a higher rice or refined grain intake was associated with body weight or with an increased risk of body weight gain [15,38]. A study in Japan suggested that a rice intake of 220 and 380 g/day reduced the risks of incident elevated BP (HRs: 0.79, 95% CI: 0.66–0.94) [39]. Usual diet habits vary depending on demography, culture, ethnicity, geographical locations, urbanization and long-term diet practice. It is necessary to understand the role of whole-grain intake on the context of ethnicity-specific dietary patterns. For example, the Western diet is characterized by increased intakes of animal proteins, saturated fatty acids, refined grains, sugar, salt, and processed foods and reduced intakes of vegetables and fruits. Given the nature of an observational study, the existing residual confounding may influence the observed effect magnitude.

Linking earlier intakes of whole and coarse grains from the CHNS 2011 to the risks of each CMF in 2015, the prospective nature of our study reduces the possibility of reverse causality to some extent. However, our study has limitations. First, without collecting baseline blood samples, those with abnormal biomarker data cannot be excluded in our study, which limited our ability to detect associations. Second, the use of consecutive 3 d 24 h dietary recalls may underestimate the intakes of episodically consumed whole grains and coarse grains. However, a relatively valid estimate of usual diet from the average intake over three days was found in earlier research using the CHNS [40]. Given the relatively small sample size and lack of baseline biochemistry data, prospective studies that overcome the above limitations would contribute to understanding this association.

In conclusion, given the significant nutrient profiles of whole grains and coarse grains, only the intake of whole grains based-on the Western definition was negatively associated with the risks of elevated serum LDL-C, and overweight and obesity.

## Figures and Tables

**Table 1 nutrients-14-02109-t001:** Baseline characteristics of participants across the levels of whole grain intake in 2011, CHNS.

	Total (*n* = 4706)	Non-Consumer (*n* = 3979)	T1 (*n* = 240)	T2 (*n* = 245)	T3 (*n* = 242)	*p* ^d^
Whole grains (g/day)	-	-	16.67(10.00,17.86)	33.33(33.33,33.33)	80.00(63.33,115.38)	<0.001
Gender, *n*(%) ^b^						0.044
Men	2108(44.79)	1800(45.24)	98(40.83)	105(42.86)	105(43.39)	
Women	2598(55.21)	2179(54.76)	142(59.17)	140(57.14)	137(56.61)	
Age, *n*(%) ^b^						0.279
18–44 years	1648(35.02)	1408(35.39)	87(36.25)	89(36.33)	64(26.45)	
45–64 years	2407(51.15)	2034(51.12)	116(48.33)	126(51.43)	131(54.13)	
≥65 years	651(13.83)	537(13.50)	37(15.42)	30(12.24)	47(19.42)	
Income level, *n*(%) ^b^						0.484
Low	1568(33.32)	1320(33.17)	82(34.17)	82(33.47)	84(34.71)	
Medium	1569(33.34)	1345(33.80)	81(33.75)	65(26.53)	78(32.23)	
High	1569(33.34)	1314(33.02)	77(32.08)	98(40.00)	80(33.06)	
Education, *n*(%) ^b^						0.008
<Primary school	1808(38.42)	1530(38.45)	77(32.08)	88(35.92)	113(46.69)	
Primary school	1568(33.32)	1312(32.97)	83(34.58)	96(39.18)	77(31.82)	
>Primary school	1330(28.26)	1137(28.58)	80(33.33)	61(24.90)	52(21.49)	
Urbanicity index, *n*(%) ^b^						<0.001
Low	1556(33.06)	1312(32.97)	58(24.17)	71(28.98)	115(47.52)	
Medium	1584(33.66)	1381(34.71)	72(30.00)	72(29.39)	59(24.38)	
High	1566(33.28)	1286(32.32)	110(45.83)	102(41.63)	68(28.10)	
Smoking, *n*(%) ^b^	1400(29.75)	1192(29.96)	63(26.25)	70(28.57)	75(30.99)	0.608
Drinking, *n*(%) ^b^	1616(34.34)	1366(34.33)	93(38.75)	74(30.20)	83(34.30)	0.269
Physical activity (MET hours/week) ^c^	221.06 ± 16.38	223.28 ± 2.99	186.92 ± 12.19	207.95 ± 12.07	231.75 ± 12.15	0.016
Dietary intake						
Total grains (g/day) ^c^	395.89 ± 17.96	393.77 ± 2.85	366.65 ± 11.61	394.70 ± 11.49	460.96 ± 11.59	<0.001
Refined grains (g/day) ^c^	377.14 ± 16.91	383.51 ± 2.68	338.80 ± 10.93	347.85 ± 10.81	340.19 ± 10.91	<0.001
Tuber (g/day) ^c^	30.82 ± 5.97	31.19 ± 0.95	27.07 ± 3.86	29.21 ± 3.82	30.13 ± 3.85	0.727
Red meat (g/day) ^c^	95.43 ± 7.72	97.17 ± 1.23	99.99 ± 4.99	88.86 ± 4.94	69.01 ± 4.98	<0.001
Poultry (g/day) ^c^	19.33 ± 3.99	19.55 ± 0.63	21.88 ± 2.58	18.48 ± 2.55	13.94 ± 2.57	0.134
Fish (g/day) ^c^	33.12 ± 5.42	33.17 ± 0.86	30.57 ± 3.50	33.47 ± 3.47	34.49 ± 3.50	0.874
Vegetables and fruits (g/day) ^c^	394.15 ± 20.32	397.43 ± 3.22	385.56 ± 13.13	380.87 ± 13.00	362.24 ± 13.11	0.038
Cooking oil (g/day) ^c^	42.90 ± 3.06	43.43 ± 0.49	44.93 ± 1.98	39.86 ± 1.96	35.26 ± 1.98	<0.001
Sodium (mg/day) ^c^	5542.00 ± 919.38	5633.49 ± 145.86	5212.99 ± 594.26	5190.31 ± 588.02	4720.02 ± 593.15	0.395
Total energy (kcal/day) ^c^	2119.00 ± 62.47	2108.62 ± 11.42	1999.11 ± 46.49	2211.06 ± 46.02	2315.39 ± 46.35	<0.001

Abbreviation: T = tertile; CMFs = cardiometabolic factors; MET = metabolic equivalent. ^b^ Data are number of participants (%). ^c^ Mean ± standard error (all such values). Adjusted by age for total energy intake and physical activity and adjusted by age and total energy intake for other food groups. ^d^ Chi-square tests for categorical variables and general linear models for continuous variables to test statistical significance of differences among groups.

**Table 2 nutrients-14-02109-t002:** Prevalence of CMFs in 2015 among Chinese adults across the baseline levels of whole grain intake.

	Total (*n* = 4706)	Non-Consumer (*n* = 3979)	Consumers			*p*-Trend ^a^
			T1 (*n* = 240)	T2 (*n* = 245)	T3 (*n* = 242)	
CMF cluster, *n*(%)	3629(78.43)	3065(78.21)	184(81.78)	195(80.91)	185(76.45)	0.819
Abdominal obesity, *n*(%)	2307(49.86)	1913(48.81)	124(55.11)	133(55.19)	137(56.61)	0.001 *
Overweight, *n*(%)	2167(46.83)	1821(46.47)	114(50.67)	126(52.28)	106(43.80)	0.633
Elevated BP, *n*(%)	2226(48.11)	1887(48.15)	99(44.00)	115(47.72)	125(51.65)	0.589
Elevated FBG, *n*(%)	1385(29.93)	1180(30.11)	74(32.89)	65(26.97)	66(27.27)	0.277
Insulin resistance, *n*(%)	653(14.11)	541(13.80)	41(18.22)	39(16.18)	32(13.22)	0.498
LR factors, *n*(%)	2782(60.13)	2360(60.22)	143(63.56)	146(60.58)	133(54.96)	0.297
Elevated TG, *n*(%)	1112(24.03)	953(24.32)	59(26.22)	54(22.41)	46(19.01)	0.088
Reduced HDL-C, *n*(%)	1682(36.35)	1394(35.57)	89(39.56)	104(43.15)	95(39.26)	0.019 *
Elevated LDL-C, *n*(%)	1644(35.53)	1408(35.93)	88(39.11)	82(34.02)	66(27.27)	0.022 *

Abbreviation: T = tertile; CMFs = cardiometabolic factors; BP = blood pressure; FBG = fasting blood glucose; LR factors = lipid-related risk factors; TG = triglycerides; HDL-C = high-density lipoprotein cholesterol; LDL-C = low-density lipoprotein cholesterol; HOMA-IR = fasting insulin (mU/mL) × fasting blood glucose (mmol/L)/22.5. ^a^ Significance for the prevalence of each CMF was determined by using Chi-square test (* *p* < 0.05), and by using the Cochran–Mantel–Haenszel test to test for trends across tertiles of whole grain intake.

**Table 3 nutrients-14-02109-t003:** Associations of baseline intakes of whole grains and coarse grains with each CMF among Chinese adults, CHNS.

	Abdominal Obesity	Overweight and Obesity	LR Factors	Elevated BP	Elevated FBG	Insulin Resistance	Elevated TG	Reduced HDL-C	Elevated LDL-C
Whole grains ^a^									
Non-consumers	Reference	Reference	Reference	Reference	Reference	Reference	Reference	Reference	Reference
T1	1.09(0.82, 1.45)	1.09(0.83, 1.43)	1.05(0.79, 1.40)	0.84(0.64, 1.12)	1.15(0.86, 1.53)	1.28(0.89, 1.83)	1.08(0.79, 1.48)	1.06(0.80, 1.42)	1.10(0.83, 1.46)
T2	1.13(0.85, 1.49)	1.17(0.90, 1.53)	0.96(0.73, 1.27)	0.92(0.70, 1.21)	0.84(0.62, 1.13)	1.14(0.79, 1.64)	0.92(0.66, 1.26)	1.35(1.02, 1.78) *	0.88(0.67, 1.17)
T3	1.08(0.79, 1.47)	0.78(0.58, 1.05)	0.81(0.60, 1.09)	0.90(0.66, 1.22)	0.83(0.60, 1.15)	0.90(0.59, 1.40)	0.77(0.53, 1.13)	1.22(0.89, 1.68)	0.64(0.46, 0.88) *
*p*-trend ^b^	0.478	0.291	0.291	0.342	0.204	0.967	0.189	0.074	0.009
Coarse grains ^c^									
Non-consumers	Reference	Reference	Reference	Reference	Reference	Reference	Reference	Reference	Reference
T1	0.98(0.74, 1.30)	1.01(0.79, 1.28)	0.94(0.72, 1.23)	0.97(0.79, 1.20)	0.88(0.70, 1.10)	1.00(0.80, 1.26)	0.89(0.66, 1.21)	0.81(0.63, 1.05)	0.84(0.68, 1.05)
T2	0.96(0.67, 1.36)	1.16(0.86, 1.56)	0.83(0.60, 1.15)	1.15(0.88, 1.49)	0.95(0.72, 1.25)	0.98(0.74, 1.29)	0.97(0.67, 1.41)	0.94(0.69, 1.29)	0.91(0.70, 1.18)
T3	0.76(0.44, 1.31)	1.33(0.85, 2.08)	1.22(0.75, 1.99)	0.93(0.64, 1.36)	0.91(0.60, 1.37)	0.93(0.61, 1.40)	1.11(0.64, 1.91)	0.84(0.52, 1.36)	0.83(0.56, 1.23)
*p*-trend ^b^	0.363	0.189	0.673	0.920	0.617	0.722	0.772	0.483	0.328

Abbreviation: T = tertile (only for consumers); CMFs = cardiometabolic factors; TG = triglycerides; BP = blood pressure, FBG = fasting blood glucose, LR factors = lipid-related risk factors; TG = triglycerides; HDL-C = high-density lipoprotein cholesterol; LDL-C = low-density lipoprotein cholesterol. ^a^ Adjusted for gender, education level, urbanicity index, physical activity, intakes of total energy, red meat, cooking oil, and refined grains. ^b^ We calculated the *p*-trend by assigning median values to tertiles of whole grains and entered this variable as a continuous term in the regression models. ^c^ Adjusted for age, education level, urbanicity index, smoking status, intakes of total energy, tuber, red meat, poultry, fish, vegetables and fruits, cooking oil, sodium, and other grains. * *p* < 0.05.

**Table 4 nutrients-14-02109-t004:** OR (95% CI) of CMFs in relation to intakes of whole grains and coarse grains in Chinese adults, CHNS.

Subgroups	Abdominal Obesity	Overweight and Obesity	LR Factors	Elevated BP	Elevated FG	Insulin Resistance	Elevated TG	Reduced HDL-C	Elevated LDL-C
Whole grains ^a^									
Non-consumers	Reference	Reference	Reference	Reference	Reference	Reference	Reference	Reference	Reference
0.01–≤50.00 g/day	1.15(0.94, 1.40)	1.16(0.95, 1.42)	1.01(0.82, 1.23)	0.90(0.74, 1.11)	1.00(0.80, 1.24)	1.23(0.94, 1.61)	0.99(0.78, 1.25)	1.18(0.97, 1.45)	1.15(0.94, 1.40)
50.01–≤150.00 g/day	1.04(0.77, 1.41)	0.73(0.54, 0.99) *	0.86(0.64, 1.17)	1.02(0.75, 1.38)	0.90(0.65, 1.26)	0.89(0.57, 1.39)	0.77(0.53, 1.12)	1.22(0.90, 1.65)	1.04(0.77, 1.41)
>150.00 g/day	1.19(0.62, 2.30)	1.01(0.53, 1.93)	0.54(0.29, 1.03)	0.80(0.42, 1.52)	0.58(0.27, 1.25)	0.86(0.33, 2.19)	0.68(0.29, 1.57)	0.73(0.37, 1.43)	1.19(0.62, 2.30)
*p*-trend ^c^	0.400	0.625	0.069	0.476	0.183	0.935	0.190	0.796	0.400
Ciarse grains ^b^									
Non-consumers	Reference	Reference	Reference	Reference	Reference	Reference	Reference	Reference	Reference
0.01–≤50.00 g/day	1.08(0.87, 1.36)	0.89(0.69, 1.14)	1.00(0.82, 1.21)	0.91(0.74, 1.11)	0.95(0.77, 1.18)	0.91(0.69, 1.20)	0.90(0.71, 1.14)	0.88(0.72, 1.07)	1.08(0.87, 1.36)
50.01–≤150.00 g/day	1.18(0.81, 1.72)	0.91(0.60, 1.37)	1.06(0.77, 1.47)	1.06(0.76, 1.47)	0.95(0.67, 1.34)	0.94(0.59, 1.48)	0.92(0.61, 1.38)	0.81(0.58, 1.14)	1.18(0.81, 1.72)
>150.00 g/day	1.53(0.62, 3.79)	0.80(0.30, 2.13)	0.88(0.41, 1.91)	1.30(0.59, 2.85)	0.71(0.31, 1.63)	0.84(0.28, 2.49)	1.42(0.55, 3.64)	0.83(0.38, 1.81)	1.53(0.62, 3.79)
*p*-trend ^c^	0.316	0.540	0.988	0.743	0.527	0.670	0.971	0.300	0.316

Abbreviation: CMFs = cardiometabolic factors; TG = triglycerides; BP = blood pressure, FBG = fasting blood glucose, LR factors = lipid-related risk factors; TG = triglycerides; HDL-C = high-density lipoprotein cholesterol; LDL-C = low-density lipoprotein cholesterol. ^a^ Adjusted for gender, education level, urbanicity index, physical activity, intakes of total energy, red meat, cooking oil, and refined grains. ^b^ Adjusted for age, education level, urbanicity index, smoking status, intakes of total energy, tuber, red meat, poultry, fish, vegetables and fruits, cooking oil, sodium, and other grains. ^c^ We calculated the *p* trend by assigning median values to tertiles of whole grains and entered this variable as a continuous term in the regression models. * *p* < 0.05.

**Table 5 nutrients-14-02109-t005:** Nutrients of coarse grains and whole grains ^a^.

Nutrients	Coarse Grains	Whole Grains	*p* ^b^
Energy (kcal/100.00 g) ^a^	207.42 ± 2.59	280.40 ± 3.28	<0.001
Protein (g/100.00 g) ^a^	5.26 ± 0.10	9.78 ± 0.12	<0.001
Fat (g/100.00 g) ^a^	1.89 ± 0.03	2.70 ± 0.04	<0.001
Carbohydrate (g/100.00 g) ^a^	42.31 ± 0.54	54.19 ± 0.69	<0.001
Fiber (g/100.00 g) ^a^	1.18 ± 0.04	6.07 ± 0.05	<0.001
Thiamin (mg/100.00 g) ^a^	0.15 ± 0.01	0.41 ± 0.01	<0.001
Riboflav (mg/100.00 g) ^a^	0.08 ± 0.00	0.13 ± 0.00	<0.001
Niacin (mg/100.00 g) ^a^	1.10 ± 0.01	2.59 ± 0.02	<0.001
Vitamin E (mg/100.00 g) ^a^	1.59 ± 0.06	3.65 ± 0.07	<0.001
Potassium (mg/100.00 g) ^a^	144.52 ± 3.68	291.38 ± 4.67	<0.001
Sodium (mg/100.00 g) ^a^	3.46 ± 0.12	3.99 ± 0.15	0.006
Calcium (mg/100.00 g) ^a^	25.59 ± 0.40	26.71 ± 0.51	0.083
Phosphorus (mg/100.00 g) ^a^	121.77 ± 2.85	255.17 ± 3.62	<0.001
Magnesium (mg/100.00 g) ^a^	68.67 ± 0.98	74.35 ± 1.25	<0.001
Iron (mg/100.00 g) ^a^	2.73 ± 0.04	3.04 ± 0.05	<0.001
Manganese (mg/100.00 g) ^a^	0.45 ± 0.04	1.62 ± 0.05	<0.001
Zinc (mg/100.00 g) ^a^	1.08 ± 0.05	2.66 ± 0.07	<0.001
Cuprum (mg/100.00 g) ^a^	0.27 ± 0.02	0.50 ± 0.02	<0.001
Selenium (μg/100.00 g) ^a^	2.56 ± 0.15	5.76 ± 0.19	<0.001

^a^ Data expressed as mean ± standard error; ^b^ statistical significance for the difference of nutrient profiles of coarse grains and whole grains were tested using general linear regression.

## Data Availability

Data sharing is not applicable to this article.

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
