# Peer review of "Differential Associations of Intakes of Whole Grains and Coarse Grains with Risks of Cardiometabolic Factors among Adults in China"

_nutrients, 2022, doi:10.3390/nu14102109_

Round 1

Reviewer 1 Report

To the authors

This manuscript is reporting association between whole grain intake and cardiometabolic risk factors. The authors examined the associations using two different definitions of “whole grains.” They reported that higher intake of whole grains used in this manuscript was associated with decreased risks of elevated LDL-C and overweight/obesity. This manuscript may contribute for the readers to understand the role of whole grains in cardiometabolic risks. There are some comments.

  1. The authors used two definitions of whole grains. But it is confusing for me. Two definitions named as “whole grain” and “coarse grains”. Whole grains were defined according to Western way and coarse grains was defined according to traditional way. They should explain their definition and their technical terms in detail carefully.
  2. Methods: As mentioned above, the section of assessment of intakes of whole grains and coarse grains should be explain in an easy-to-understand manner.
  3. Methods: The authors seemed to use crude consumption of whole grains. I wonder whether high consumption of whole grains or coarse grains was associated with high energy consumption, and this can affect their results. They can consider energy-adjusted whole grain consumption.
  4. Methods: The authors may not explain the methods to assess nutrients-profiles of coarse grains and whole grains. If so, please explain it.
  5. Line 186: “differentacross” may read “different across”.
  6. Results: Results of odds ratio for over weight /obesity in Table 4 seemed to be U shape. Are the results confounded with energy intake?
  7. Discussion: In lines 234-249, the authors discussed differences between previous reports and their result. I think that the effects of usual diet of each study ethnicity are important for understanding the role of whole grains. The author can discuss these differences considering based on diet habit of each ethnicity.
  8. Table 3 and 4: The order of whole grains and coarse grains should be same way.
  9. References: I think that the paper of Kissock KR (Adv Nutr 2021 Jun 1;12(3):693-707) is useful.

Author Response

Dear Professor,
      Thank you so much for your constructive comments. We have revised the manuscript based on your suggestions. We marked the changes in yellow in the revised manuscript. Please let know if there is any problems with our responses and revison. We really appreciate your great help and supports.

 Zhihong Wang

Reviewer 2 Report

In this study, the Authors classified whole grains into two classes and investigated its associations with risks of major CMFs among in Chinese population using data from CHNS 2011 and CHNS 2015.

The aim of this study is interesting, I have the following suggestions/comments:

1. Please check the structure of the below sentences.

"Cardiovascular diseases (CVDs) produce immense health and economic burdens in globally"

"However, the results remained inconsistent nowadays."

"refined grain, results remained controversy many studies showed refined grain intake was"

"There is lack of studies on the association of whole grains intake and CMFs in China"

2. Put reference for below sentence. 

"However, the results remained inconsistent nowadays."

3. Are there any existing classifications of grains, if yes please briefly mention that as well and discuss the drawbacks of those classifications if any.

4. provide the full form of any abbreviation in ist first mention, e.g., CHNS

5. THe study was approved by the University of North Carolina at Chapel Hill IRB, however, none of the investigators are from this institution. Please look into this.

6. Please list the example for the grains that were classified as Coarse grains.

7. Please see the below line, please mention if its a brand. Currently looks like its a name of the worker that measured the weight and height

"Using BC601, TANITA the trained"

8. Please look at the below line. It seems contradictory with regard to wheat and their products intake.

"In our analysis, we found that higher intake of rice, wheat and their products as well as refined grains would not help to reduce the OR of overweight and obesity, but the significance only were observed from the intake of wheat and their products."

9. Please replace reference 1 with the latest GBD burden paper.

"Global burden of 369 diseases and injuries in 204 countries and territories, 1990–2019: a systematic analysis for the Global Burden of Disease Study 2019"

10. with regards to "adjusting for covariates as needed" please add details on how the covariates that were adjusted in the final model were identified.

Author Response

(The authors gave the same response as above.)

Round 2

Reviewer 1 Report

To the authors

All of my criticisms have been addressed.

Reviewer 2 Report

Thank you for addressing the concerns and including the suggestions.